# Miktoarm Star Copolymers Prepared by Transformation from Enhanced Spin Capturing Polymerization to Nitroxide-Mediated Polymerization (ESCP-*Ŧ*-NMP) toward Nanomaterials

**DOI:** 10.3390/nano11092392

**Published:** 2021-09-14

**Authors:** Tzu-Yao Lin, Cheng-Wei Tu, Junko Aimi, Yu-Wen Huang, Tongsai Jamnongkan, Han-Yu Hsueh, Kun-Yi Andrew Lin, Chih-Feng Huang

**Affiliations:** 1Department of Chemical Engineering, i-Center for Advanced Science and Technology (iCAST), National Chung Hsing University, Taichung 40227, Taiwan; s05310141@gmail.com (T.-Y.L.); enjoy1995427@gmail.com (Y.-W.H.); 2Industrial Technology Research Institute, Chutung, Hsinchu 31057, Taiwan; CWTu@itri.org.tw; 3Research Center for Functional Materials, National Institute for Materials Science, 1-2-1 Sengen, Tsukuba 305-0047, Ibaraki, Japan; AIMI.Junko@nims.go.jp; 4Department of Fundamental Science and Physical Education, Faculty of Science at Sriracha, Kasetsart University, Chonburi 20230, Thailand; jamnongkan.t@ku.ac.th; 5Department of Materials Science and Engineering, National Chung Hsing University, Taichung 40227, Taiwan; hyhsueh@nchu.edu.tw; 6Department of Environmental Engineering, Innovation and Development Center of Sustainable Agriculture, Research Center of Sustainable Energy and Nanotechnology, iCAST, National Chung Hsing University, Taichung 40227, Taiwan

**Keywords:** ESCP, NMP, miktoarm star copolymers, polystyrene, poly(*tert*-butyl acrylate)

## Abstract

Reversible-deactivation radical polymerization (RDRP) serves as a powerful tool nowadays for the preparations of unique linear and non-linear macromolecules. In this study, enhanced spin capturing polymerizations (ESCPs) of styrene (St) and *tert*-butyl acrylate (tBA) monomers were, respectively, conducted in the presence of difunctional (1Z,1′Z)-1,1′-(1,4-phenylene) bis (*N*-*tert*-butylmethanimine oxide) (PBBN) nitrone. Four-arm (PSt)_4_ and (PtBA)_4_ star macroinitiators (MIs) can be afforded. By correspondingly switching the second monomer (i.e., tBA and St), miktoarm star copolymers (μ-stars) of (PSt)_2_-μ-(PtBA-*b*-PSt)_2_ and (PtBA)_2_-μ-(PSt-*b*-PtBA)_2_) were thus obtained. We further conducted hydrolysis of the PtBA segments to PAA (i.e., poly(acrylic acid)) in μ-stars to afford amphiphilic μ-stars of (PSt)_2_-μ-(PAA-*b*-PSt)_2_ and (PAA)_2_-μ-(PSt-*b*-PAA)_2_. We investigated each polymerization step and characterized the obtained two sets of “sequence-isomeric” μ-stars by FT-IR, ^1^H NMR, differential scanning calorimeter (DSC), and thermogravimetric analysis (TGA). Interestingly, we identified their physical property differences in the case of amphiphilic μ-stars by water contact angle (WCA) and atomic force microscopy (AFM) measurements. We thus proposed two microstructures caused by the difference of polymer chain sequences. Through this polymerization transformation (*Ŧ*) approach (i.e., ESCP-*Ŧ*-NMP), we demonstrated an interesting and facile strategy for the preparations of μ-stars with adjustable/switchable interior and exterior polymer structures toward the preparations of various nanomaterials.

## 1. Introduction

With the development of reversible-deactivation radical polymerizations (RDRPs) [1,2], the controlled/living polymerization sort can be facilely applied to a wide range of functional vinyl monomers and we can obtain predictable molecular weights (MWs) and low polydispersity (PDI). Most well-known RDRP techniques include nitroxide-mediated polymerization (NMP) [3,4,5], atom transfer radical polymerization (ATRP) [6,7,8], reversible addition–fragmentation chain transfer (RAFT) polymerization [9,10,11,12], and numerous other methods [13,14,15,16,17]. Nowadays, these powerful RDRPs significantly enrich the manipulations of polymer architectures [18,19,20,21,22,23], compositions [24,25], and multi-functionalities [26,27], as well as in the preparations of nanomaterials [28,29,30].

Among RDRP techniques, NMP is recognized as one of the most adaptable techniques for industrial applications due to its simple reaction condition. Previously, the Jérôme group [31,32] and the Barner–Kowollik group [33,34] demonstrated an innovative method called “enhanced spin capturing polymerization” (ESCP), which was basically extended from NMP and can also be broadly classified in the same family. The newly developed ESCP is conducted in conventional free radical polymerization (FRP) conditions but is necessary for adding nitrone compound. During polymerization, intermediates can be effectively formed via a cross-trapping reaction between propagating macroradicals and nitrones. Then one intermediate macroradical can quickly undergo cross-coupling with another propagating macroradical. Accordingly, a new alkoxyamine moiety can result, which can further conduct NMP from the middle of polymer chains. Namely, the transformation from ESCP to NMP (i.e., ESCP-*Ŧ*-NMP) can be facilely attained. Through ESCP of monomer A in the presence of a difunctional nitrone, a four-arm star macroinitiator (i.e., (PA)_4_ MI), having both inactive and active arms, can be afforded. More interestingly, NMP of monomer B from the (PA)_4_ MI can be further conducted to obtain unique μ-stars (i.e., (PA)_2_-μ-(PB-*b*-PA)_2_). The beauty of this convergent approach is that it can be used to change the polymerization sequence of monomers. Therefore, a “sequence-isomer” of (PB)_2_-μ-(PA-*b*-PB)_2_ can be alternatively attained.

Combining with RDRPs and several other living polymerization techniques [35,36,37,38,39], sequential polymerization approaches via single chemistry or polymerization transformation (*Ŧ*) provided a diverse way to create novel and unique macromolecules [40,41,42], including typical BCPs, soft-hard segmented BCPs, star block copolymers (sBCPs), and miktoarm star copolymers (μ-stars). The polymer chain topology of BCPs typically belongs to the analog of linear type copolymers and comprises at least two chemically linked (im)miscible polymer segments. Accordingly, BCPs can spontaneously conduct interesting microphase separation behaviors to fabricate nanomaterials. By further going to non-linear type “segmented” copolymers, the design of fascinating polymer architectures, such as (A-*b*-B)_n_ sBCPs and A_2_B/ABC/A_3_B/ABCD μ-stars, has received significant attention due to their specific and unique physical properties [43,44,45,46,47,48]. Several examples studied the effectiveness of sBCPs on the self-assembly behaviors and further addressed the critical effect of “entropy penalty” coming from the nature of complex polymer architectures [49,50].

In an attempt to design unique and specific non-linear copolymers, we herein combine a sequential polymerization transformation of ESCP-*Ŧ*-NMP. As shown in route I in Figure 1, azobisisobutyronitrile (AIBN) can be thermally activated to generate radicals following the addition of styrene (St) monomer. The single addition of a primary monomer radical can further undergo chain propagations. During polymerizations, the propagating PSt• species can be captured by the difunctional PBBN nitrone in a four-fold molar ratio. Due to the ESCP mechanism being based on a methodology of convergent approach, a four-arm (PSt_n_)_4_ star homopolymer (homo-star) can be accordingly attained. With *tert*-butyl acrylate (tBA) as a monomer, as shown in route II in Figure 1, a four-arm (PtBA_x_)_4_ star homopolymer (homo-star) can also be attained via the ESCP mechanism. It is more important and interesting that the resulting homo-stars both possess two thermally reversible alkoxyamine moieties that can be subsequently extended to proceed with nitroxide-mediated radical polymerization (NMP). With different polymerization sequences of St and tBA, namely, “sequence-isomers” of miktoarm star copolymers can be facilely obtained that might have different physical properties. We characterized the copolymers using Fourier-transform infrared (FT-IR) spectroscopy, differential scanning calorimetry (DSC), thermogravimetric analysis (TGA), water contact angle (WCA) measurements, and atomic force microscope (AFM).

## 2. Materials and Methods

### 2.1. Materials

Terephthalaldehyde (99%), 2-methyl-2-nitro-1-propanol (MNP, 98%), *N*-(*tert*-butyl)hydroxylamine acetate (97%), triethylamine (99%), azobisisobutyronitrile (AIBN, 99%), styrene (St, 99.5%), *tert*-butyl acrylate (tBA, 98%), tributyltin hydride (TBTH, 98%), trifluoroacetic acid (TFA, 99%), and 1-methoxy-2-propyl acetate (MPA, 99.5%) were purchased from Sigma–Aldrich (St. Louis, MI, USA). Monomers and solvents were purified before use.

### 2.2. Characterization

The monomer conversions were analyzed by a Hewlett–Packard gas chromatograph (GC) set (a HP 5890 ser. II FID detector and a CD-5 column). Gel permeation chromatography (GPC) with two PSS SDV columns (linear S and 100 Å) and an RI detector at 35 °C (eluent: tetrahydrofuran (THF); flow rate: 1 mL/min) were used. Characterization of number and weight average molecular weights (i.e., *M*_n_ and *M*_w_) and polydispersity (PDI = *M*_w_/*M*_n_) was calculated from a polystyrene calibration line. Functional groups of samples were analyzed with the FT-IR instrument of PerkinElmer Spectrum One. Each sample was drop-casted on a KBr disk and dried in a vacuum for a few hours. A Bruker 400 NMR spectrometer was utilized to record ^1^H NMR spectra using CDCl_3_ (chemical shift (δ) = 7.26 ppm) for calibration. Seiko 6220 differential scanning calorimeter (DSC) was utilized to detect the glass transition temperatures (*T*_g_) of the (co)polymers under N_2_ (first run up to 180 °C to remove thermal history and quenching; the second run was ramped with 20 °C/min from −10 to 190 °C). TA Instruments Q50 thermogravimetric analysis (TGA) was used to trace the thermal stability of (co)polymers loaded on a platinum holder (runs were ramped with 20 °C/min in the range of 25–700 °C under N_2(g)_). Water contact angles (WCAs) were measured with the KRÜSS G10 instrument with a neutral DI water droplet (ca. 5 μL). Seiko SPA400 atomic force microscopy (AFM) was used (1 wt% of polymer solutions in 1-methoxy-2-propyl acetate (MPA) were drop-casted onto wafers).

### 2.3. Synthesis of (1Z,1′Z)-1,1′-(1,4-Phenylene) bis (N-Tert-Butylmethanimine Oxide) (PBBN)

As shown in Appendix A, a round-bottomed flask was charged with terephthalaldehyde (1.02 g, 7.46 mmol), *N*-(*tert*-butyl) hydroxylamine acetate (2.41 g, 16.4 mmol), excess magnesium sulfate anhydrous (MgSO_4_), and chloroform (50.0 mL). Triethylamine (2.31 mL, 16.4 mmol) was dropwise added to the flask and stirred at room temperature for 3 days. After the reaction was completed, the solution was concentrated and added to D. I. water. The mixture was extracted with ethyl acetate. The organic layers were collected and dried over MgSO_4_. The crude product was recrystallized by EtOH and a pale yellow PBBN nitrone compound was acquired (yield 75.1%). FT-IR (ν = cm^−1^): 2969 (m), 1573 (w), 1414 (m), 1361 (s), 1261 (s), 1196 (s), 1122 (s), 1019 (s), 803 (s). ^1^H NMR (400 MHz, CDCl_3_, δ = ppm): 1.62 (s, 18H), 7.58 (s, 2H), 8.32 (s, 4H). Anal. calc. for C_16_H_24_N_2_O_2_: C, 69.53; H, 8.75; N, 10.14; O, 118. Found: C, 68.73; H, 8.26; N, 9.51. FT-IR and ^1^H NMR spectra were shown in Appendix A.

### 2.4. ESCPs of St and tBA and Thermolysis of the (PSt)_4_ Homo-Star

For the ESCP of St (i.e., route I in Figure 1), PBBN (36.2 mg, 0.131 mmol), St (6.05 mL, 52.1 mmol), AIBN (21.5 mg, 0.131 mmol), and a few drops of anisole were mixed in a Schlenk flask. The reaction flask was sealed and degassed through four freeze/pump/thaw cycles. The flask was placed in an oil bath at 80 °C and the monomer conversion was traced by GC. The molecular weight of the product was estimated by using GPC. To stop the reaction, the flask was cooled by an ice bath and exposed to air. The solution was diluted with THF and precipitation in MeOH. The powder was collected and dried in a vacuum oven to remove residual solvent. White powder of 4-arm PSt homopolymer (i.e., (PSt)_4_ homo-star: yield 36.5%, *M*_n_ = 17760, PDI = 1.53) was used. For the ESCP of tBA (i.e., route II in Figure 1), the procedures were similar to route I and afforded pale yellow viscous liquid of 4-arm PtBA homopolymer (i.e., (PtBA)_4_ homo-star: yield 32.5%, *M*_n_ = 13560, PDI = 1.55).

For thermolysis of (PSt)_4_ homo-star, a Schlenk flask was charged with P(St)_4_ (0.211 g, 10.1 μmol), TBTH (623 μL, 2.01 mmol), and DMF (25.2 mL). The mixture was refluxed overnight. The molecular weight changes through the selective cleavage reaction were monitored by using GPC.

### 2.5. Synthesis of Miktoarm Star (μ-Star) Copolymers via Nitroxide-Mediated Chain Extensions of Homo-Stars and Hydrolysis of PtBA Segments

For the nitroxide-mediated chain extensions of homo-stars (i.e., route III(i) in Scheme 2), (PSt)_4_ homo-star (0.82 g, 0.03 mmol) macroinitiator, tBA (0.351 mL, 2.42 mmol), and anisole were added to a Schlenk flask. The degassing method was similar to the above-mentioned procedures for removing oxygen. The flask was placed in a 125 °C oil bath and the monomer conversion was traced by GC. The molecular weight evolution was monitored by GPC. After a desired period, the reaction was quenched and opened to air. The mixture was diluted with DCM and precipitated in MeOH: H_2_O = 8/2 (*v*/*v*). Yellowish white powder of (PSt)_2_-μ-(PtBA-*b*-PSt)_2_ miktoarm star (μ-star) copolymer was acquired (yield 71%). With (PtBA)_4_ homo-star as a macroinitiator and St as a monomer (i.e., route IV(i) in Scheme 2); similarily, yellowish-white powder of (PtBA)_2_-μ-(PSt-*b*-PtBA)_2_ μ-star copolymer was acquired (yield 54.9%).

An example of hydrolysis: (PtBA)_2_-μ-(PSt-*b*-PtBA)_2_ μ-star (0.721 g, 60.1 μmol), trifluoroacetic acid (0.502 mL, 6.11 mmol), and DCM (15.1 mL) were mixed and kept overnight at room temperature. After the reaction was completed, the mixture was purified by dialysis membrane (MWCO = 3500) for three days using DMSO: DI water = 5/95 (*v*/*v*). The purified solution was freeze-dried to remove water. White powder of (PAA)_2_-μ-(PSt-*b*-PAA)_2_ μ-star was obtained (yield 57.8%). To obtain (PSt)_2_-μ-(PAA-*b*-PSt)_2_, similar procedures were conducted. FT-IR spectra of the obtained μ-star copolymers before and after hydrolysis are shown in Appendix A.

## 3. Results and Discussion

Figure 1 shows the kinetic plots of ESCP of styrene (St) and *tert*-butyl acrylate (tBA) monomers (Ms) using PBBN nitrone at 80 °C (M/AIBN/PBBN = 400/1/1; [M]_0_ = 4.0 M). In Figure 1a, both cases of ESCPs (i.e., solid symbols) showed deviations in a short reaction time. In Figure 1b,c, we obtained broad molecular weight distributions (i.e., PDI > 1.5) and only slight changes in molecular weights (MWs) concerning the monomer conversions (i.e., *M*_n,PSt_ = ca. 16k and *M*_n,PtBA_ = ca. 13k). As shown in the corresponding GPC traces in Figure 2, broad mono-modal curves can be observed with only slight changes in MWs. These results indicate that the polymerizations were poorly controlled and that high PDI values were obtained. Based on the reported mechanism, an ESCP typically proceeds in a conventional free radical polymerization fashion. Namely, ESCP possesses the non-RDRP feature. These results are similar to the reported studies [33,34]. As illustrated in Figure 1, two macroinitiators of 4-arm (PSt)_4_ and (PtBA)_4_ star homopolymers (homo-stars) were thus obtained, which both possess two active arms connected by the alkoxyamine moieties to the star center. Namely, thermal dissociability of the macroinitiators can be expected.

To confirm the dissociability of NO–C linkages, we further conducted thermal cleavage of (PSt)_4_ homo-star by TBTH at 125 °C [(PSt)_4_/TBTH = 1/20; [(PSt)_4_]_0_ = 4 mM in DMF]. As illustrated in Figure 3A, the NO–PSt linkages can be thermally dissociated and generated two types of PSt macroradicals. In the presence of excess amounts of TBTH reducing agent, mixed PSt chains can be thus obtained, which are composed of linked two arms (i.e., one single (PSt)_2_ chain) and two single arms (i.e., two PSt chains). After the thermolysis reaction, as shown in Figure 3B, MWs displayed a negative evolution in GPC traces from curve b (i.e., the original (PSt)_4_: *M*_n_ = 16,120 and PDI = 1.52) to curve a (*M*_n,overall_ = 8900 and PDI = 1.34). The significant decrease in MW was reasonably ascribed to the cleavage of (PSt)_4_ homo-star, reductions of PSt macroradicals, and generations of single indivisible (PSt)_2_ (i.e., species **1**) and two cleaved PSt chains (i.e., species **2**). At the end of curve b in Figure 3B, an elevated trend might be due to the baseline deviation issue. These results rationally revealed the presence of reversible alkoxyamine moiety in the homo-stars that can be further carried out post-polymerizations.

As shown in routes III(i) and IV(i) in Figure 2, we can expect that the two dissociable NO–PSt linkages can be thermally reactivated and proceed with NMP of second monomer to afford miktoarm star (μ-star) copolymers. With different polymerization sequence sets, we can conduct chain extensions of (PSt)_4_ and (PtBA)_4_ homo-star macroinitiators (MIs) with a different second monomer in bulk (i.e., tBA and St, respectively). Figure 4 displays kinetic plots of the chain extensions of (PSt)_4_ with tBA (i.e., circle symbols) and (PtBA)_4_ with St (i.e., square symbols). As shown in Figure 4a, both chain extension cases performed a gradual consumption of monomers. Linear first-order reactions were observed in each case. An apparent faster consumption in the chain extension with tBA (*k*_app(tBA)_ = (7.34 ± 0.6) × 10^−4^ s^−1^) and a slower reaction rate in the chain extension with St (*k*_app(St)_ = (1.18 ± 0.2) × 10^−4^ s^−1^) were acquired. The significant difference (*k*_app(tBA)_/*k*_app(St)_ = ca. 6 times) might be ascribed to obvious difference of propagation rate constants of the two monomers (e.g., *k*_p(tBA)_ = ca. 10^4^ s^−1^ and *k*_p(St)_ = ca. 10^3^ s^−1^ at 125 °C) [7,51]. As shown in Figure 4b, PDI values of less than 1.7 were acquired in the conversion of less than 70%. With the conversion of ca. 90%, a high PDI value (= ca. 2.11) was observed that might be due to the occurrence of side reactions. Several side reactions during NMPs were reported, mainly including disproportionation of the dissociated alkoxyamines, cross-reactions of intramolecular/intermolecular hydrogen-transfer, and irreversible N–OC bond cleavage [5,52]. As shown in Figure 4c, however, we observed an unexpected slight decrease trend in MWs. It is reported that the more complex molecular architectures cause smaller hydrodynamic volumes in the same solution environment in comparison to those of linear polymers in a similar MW [20,53]. In the GPC analysis, we utilized linear PSt as the calibration standard. Another possible reason is that the insertion of polymer chains during NMP led to the formation of μ-star copolymers with an asymmetric molecular conformation. Thus, the MW deviations could be rationally ascribed to the synergistic effects of asymmetry and star-shaped architecture. As shown in Figure 5, the corresponding GPC traces displayed a monomodal and only slight increases in PDI values in moderate conversions. These results thus indicated the subsequent chain extensions of NMPs, proceeding in a controlled/living radical polymerization fashion. Accordingly, the unique and facile multisite polymerization transformation (*Ŧ*) from ESCP to NMP (i.e., ESCP-*Ŧ*-NMP) provides an interesting and fascinating strategy for the preparations of μ-stars with switchable/adjustable interior and exterior polymer structures. Two kinds of miktoarm star (μ-star) copolymers (i.e., (PSt)_2_-μ-(PtBA-*b*-PSt)_2_ and (PtBA)_2_-μ-(PSt-*b*-PtBA)_2_) were thus afforded.

As shown in routes III(ii) and IV(ii) in Figure 2, we further conducted hydrolysis of the PtBA segments in μ-star copolymers via TFA. Figure 6A,B represent ^1^H NMR spectra (400 MHz, CDCl_3_) of the original (PSt)_2_-μ-(PtBA-*b*-PSt)_2_ and (PtBA)_2_-μ-(PSt-*b*-PtBA)_2_ μ-star copolymers. These two spectra have similar peaks, including aromatics (i.e., peak Ar) and alkyls (i.e., peaks a–c), resulting from the chemical structures of PSt and PtBA segments. Figure 6C,D represent ^1^H NMR spectra (400 MHz, DMSO-*d*_6_) of the products that were obtained after the hydrolysis reactions were completed. These two spectra are also similar, including aromatics (i.e., peak Ar), alkyls (i.e., peaks a and b), and acid (i.e., peak c’), revealing our successful deprotection of the *tert*-butyl group (tBu); the PSt segments remained. We thus acquired (PSt)_2_-μ-(PAA-*b*-PSt)_2_ and (PAA)_2_-μ-(PSt-*b*-PAA)_2_ amphiphilic μ-star copolymers.

Figure 7 represents DSC traces of various μ-star copolymers in a range of 10–180 °C. In the samples of (PSt)_2_-μ-(PtBA-*b*-PSt)_2_ and (PtBA)_2_-μ-(PSt-*b*-PtBA)_2_ μ-star copolymers (i.e., curves a and b), two *T*_g_s were obviously observed, resulting from the PtBA segments (ca. 46 °C) and PSt segments (ca. 102 °C). These results elucidated the nature of their amorphous and immiscible behaviors. In the samples of (PSt)_2_-μ-(PAA-*b*-PSt)_2_ and (PAA)_2_-μ-(PSt-*b*-PAA)_2_ μ-star copolymers (i.e., curves c and d), we observed the disappearance of the original *T*_g_ from PtBA segments and acquired two other *T*_g_s, resulting from the PSt segments (ca. 100 °C) and PAA segments (ca. 135 °C). These results also indicated the natures of amorphous and immiscible behaviors.

The thermal stability of the μ-star copolymers and relevant homopolymers was examined by TGA and displayed in Figure 8. Linear PSt, PtBA, and PAA were shown in traces a–c. In the cases of (PSt)_2_-μ-(PtBA-*b*-PSt)_2_ and (PtBA)_2_-μ-(PSt-*b*-PtBA)_2_ μ-star copolymers (i.e., curves d and e with solid symbols), similar two-step decomposition profiles were observed. Two maximum rates (*r*_d_) were revealed at approximately 250 and 410 °C, respectively. The first step might be mainly due to pyrolysis of the pendent tBu bulky group of PtBA segments. Then the second step arose from pyrolysis of the residual main chains. The thermal stabilities, based on 5 wt% weight loss temperatures (*T*_d5%_), were similar at approximately 232 °C. The char yields were both approximately 0.5 wt%. In the cases of (PSt)_2_-μ-(PAA-*b*-PSt)_2_ and (PAA)_2_-μ-(PSt-*b*-PAA)_2_ μ-star copolymers (i.e., curves f and g with hollow symbols), two-step decomposition profiles were also observed. However, the first step showed a gradual decomposition profile in both cases. It is rationally due to continuous dehydration and pyrolysis of the pendent carboxylic acid of PAA segments. Then the second step also arose from pyrolysis of the residual main chains. *T*_d5%_s were about 220 and 165 °C, respectively. Char yields were both approximately 0.3 wt%.

The μ-stars were spin-coated on silicon wafers and the hydrophobicity was analyzed through water contact angle (WCA) measurements. As shown in Figure 9a,b, the WCAs of (PSt)_2_-μ-(PtBA-*b*-PSt)_2_ and (PtBA)_2_-μ-(PSt-*b*-PtBA)_2_ were ca. 99.9 ± 4.5° and 104.6 ± 5.6°, respectively. The results revealed their similar hydrophobicity despite the different interior and exterior sequences of PSt and PtBA chains. After hydrolysis of the pendent group of PtBA segments, we further examined WCAs vs. different holding times. As displayed in the set of Figure 9c, we initially observed a WCA of ca. 98.9 ± 5.2° in the (PSt)_2_-μ-(PAA-*b*-PSt)_2_ sample. With different holding times, we then observed a slight decrease of WCAs in a minute to ca. 83.5 ± 4.3°. The only slight decreases in WCAs revealed its stable hydrophobicity. With the nature of immiscible PSt and PAA chains, we deduced that a low wettability microstructure having PAA segments as a discontinuous phase surrounded by a continuous phase of PSt segments as was formed. As displayed in the set of Figure 9d, we initially observed a similar WCA of ca. 90.9 ± 4.7° in the (PAA)_2_-μ-(PSt-*b*-PAA)_2_ sample. With different holding times, interestingly, we then observed a significant decrease of WCAs in a minute to ca. 24.9 ± 2.3°. The WCA changes were due to the surficial swelling behaviors between the PAA segments and water droplets [54]. Accordingly, we deduced that a wettable microstructure with PSt segments as a discontinuous phase covered by a continuous phase of PAA segments was constructed. The significant different hydrophobic properties are synergy via the complex μ-star architecture and amphiphilic polymer chains.

The above two amphiphilic μ-stars were then dissolved in 1-methoxy-2-propyl acetate (MPA) (1 wt%) and drop-casted on silicon wafers to analyze the morphology by AFM. Figure 10 displays the topologies of (PSt)_2_-μ-(PAA-*b*-PSt)_2_ and (PAA)_2_-μ-(PSt-*b*-PAA)_2_ μ-stars that both indicated quite smooth coating films. The height image in Figure 10a revealed a morphology with isolated dark domains surrounded by a continuous white matrix. The corresponding phase image in Figure 10a displayed an opposite color compared to that of the height image. The isolated white domains resulted from hard PAA chains that possess a high *T*_g_ of ca. 135 °C. Thus, the continuous dark matrix was mainly composed of softer PSt chains (*T*_g_ = ca. 100 °C). From the abovementioned two images, we thus proposed a microstructure that was inserted in the middle of Figure 10a. The proposed microstructure elucidated that the interior PAA segments (i.e., blue-colored chains) served as a discontinuous phase and were surrounded by a continuous phase of PSt-rich domains (i.e., black colored chains). In the case with an alternative sequence of (PAA)_2_-μ-(PSt-*b*-PAA)_2_ μ-star, the height image in Figure 10b displayed a connected particle-like morphology. The corresponding phase image in Figure 10b showed major white and light-brown colors, indicating the surface was majorly composed of hard PAA segments. From the two images in Figure 10b, a proposed microstructure was also inserted. The proposed microstructure indicated that the interior PSt segments (i.e., black-colored chains) served as a discontinuous phase mostly covered by a continuous phase of the PAA segments (i.e., blue-colored chains). The two proposed microstructures could also be consistent with the behaviors observed in the WCA measurements. Interestingly, we first demonstrated changeable μ-star chemical structures via polymerization sequence control led to dissimilar physical properties composing different sequenced μ-stars.

## 4. Conclusions

ESCPs of St and tBA monomers with PBBN as the capturing agent were first demonstrated. We then obtained 4-arm (PSt)_4_ and (PtBA)_4_ homo-stars with two thermally reversible alkoxyamine moieties. In the presence of excess amounts of TBTH, the dissociations of NO–C linkages can be revealed by GPC traces. The 4-arm (PSt)_4_ and (PtBA)_4_ macroinitiators were subsequently proceeded by NMPs of tBA and St, respectively. We observed first-order kinetic plots but the GPC traces showed unexpected decrease trends in MWs. This might be attributed to the synergistic influences of asymmetry and star-shaped architecture. Combining kinetic data with GPC traces (i.e., a monomodal and only slight increases in PDIs), the subsequent chain extensions of NMPs were carried out in a controlled/living radical polymerization fashion. “Sequentially isomeric” μ-stars of (PSt)_2_-μ-(PtBA-*b*-PSt)_2_ and (PtBA)_2_-μ-(PSt-*b*-PtBA)_2_ were thus afforded. By further hydrolysis of the PtBA segments, another set of sequentially isomeric μ-stars of (PSt)_2_-μ-(PAA-*b*-PSt)_2_ and (PAA)_2_-μ-(PSt-*b*-PAA)_2_ can be attained. In DSC and TGA analyses, the two sets of sequentially isomeric μ-stars performed similar thermal properties. In WCA measurements of spin-coated μ-star films on wafers, we observed a similar hydrophobicity in (PSt)_2_-μ-(PtBA-*b*-PSt)_2_ and (PtBA)_2_-μ-(PSt-*b*-PtBA)_2_. Interestingly, we observed quite different wetting behaviors after hydrolysis, indicating a significant contribution from the polymer chain sequence in amphiphilic μ-stars. The spin-coated thin films of the amphiphilic μ-stars were further examined by AFM and revealed different morphologies. We thus proposed two microstructures: (i) In the case of (PSt)_2_-μ-(PAA-*b*-PSt)_2_, the interior PAA segments served as a discontinuous phase and were surrounded by a continuous phase of PSt-rich domains. (ii) In the case of (PAA)_2_-μ-(PSt-*b*-PAA)_2_, oppositely, the interior PSt segments served as a discontinuous phase mostly covered by a continuous phase of the PAA segments. In sum, the facile polymerization transformation strategy of ESCP-*Ŧ*-NMP provides an interesting and fascinating strategy for the preparations of μ-stars with switchable/adjustable interior and exterior polymer structures. We eventually acquired amphiphilic μ-stars and demonstrated their significant differences in thin film self-assembly.

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
