# Peer review of "Miktoarm Star Copolymers Prepared by Transformation from Enhanced Spin Capturing Polymerization to Nitroxide-Mediated Polymerization (ESCP-Ŧ-NMP) toward Nanomaterials"

_nanomaterials, 2021, doi:10.3390/nano11092392_

Round 1

Reviewer 1 Report

This paper presents an interesting idea:  using PBBN (a compound with dual nitrones) to produce star-type polymers.  The chemistry relies on PBBN first capturing a PS or PMA radical which converts the nitrone into a nitroxide which serves as a mediator for the nitroxide mediated polymerization to install the other arms.  While the idea is interesting, the following questions remain unclear:

1) Why do the authors believe this to be an example of RDRP?  For example, the PDI is over 1.5 and Mn is not increasing with conversion (Figure 1) for the homopolymers.  The polymerization of tBA seems to stall at less than 30% conversion, while the apparent Mn of PS seems to reach its maximum at 10% conversion then plateau.  

2) Is this data from a single run in each case?  It would be nice to see how results are affected by monomer:initiator ratios, temperature, and perhaps solvent.  There are no data tables so it's hard to tell if the results are typical or simply a single run under one set of conditions.  

3) The GPC traces in Fig 5 are poor, and appear to have been cut off before they returned to baseline.  This means that the actual PDI values and Mn values would be altered.  For example, the 0.75h in figure 5a looks like there is a much more substantial low MW tail cut off.

4) While the analyses such as WCA and TGA are a nice addition, the focus should be on the polymerization system since the introduction and abstract are clear in their emphasis of RDRP.  If the authors want to de-emphasize this feature and focus more on the properties of the hydrolyzed materials, such as the blocks, then some of the above concerns are negated.

Author Response

Response to the Reviewers’ comments (Manuscript ID: nanomaterials-1355637):

For reviewer 1:

This paper presents an interesting idea: using PBBN (a compound with dual nitrones) to produce star-type polymers. The chemistry relies on PBBN first capturing a PS or PMA radical which converts the nitrone into a nitroxide which serves as a mediator for the nitroxide mediated polymerization to install the other arms. While the idea is interesting, the following questions remain unclear.

General response: We sincerely thank the reviewer’s positive remarks. Our point-by-point responses are listed as follows.

  1. 1. Why do the authors believe this to be an example of RDRP? For example, the PDI is over 1.5 and Mn is not increasing with conversion (Figure 1) for the homopolymers.  The polymerization of tBA seems to stall at less than 30% conversion, while the apparent Mn of PS seems to reach its maximum at 10% conversion then plateau.

Response: We thank the reviewer’s important comment. Yes, ESCP does not possess the feature of controlled/living radical polymerization. We revised and addressed this point in the text. We marked the text in yellow.

  1. 2. Is this data from a single run in each case? It would be nice to see how results are affected by monomer:initiator ratios, temperature, and perhaps solvent.  There are no data tables so it's hard to tell if the results are typical or simply a single run under one set of conditions.  

Response: We thank the reviewer’s constructive comment. We added error bars in similar polymerization conditions. We marked the changes in yellow. Systematic polymerizations of ESCP-Ŧ-NMP, covering M/I, T, and solvents, are currently underway. We hope to present soon after the tough situation. 

  1. 3. The GPC traces in Fig 5 are poor, and appear to have been cut off before they returned to baseline. This means that the actual PDI values and Mn values would be altered.  For example, the 0.75h in figure 5a looks like there is a much more substantial low MW tail cut off.

Response: We thank the reviewer’s critical comment. It was due to our baseline issue. We re-estimated again and revised the trace.

  1. 4. While the analyses such as WCA and TGA are a nice addition, the focus should be on the polymerization system since the introduction and abstract are clear in their emphasis of RDRP. If the authors want to de-emphasize this feature and focus more on the properties of the hydrolyzed materials, such as the blocks, then some of the above concerns are negated.

Response: We thank the reviewer’s valuable and important comments. We revised the introduction part by shortening the detailed descriptions of RDRPs. The changes are marked in blue. 

Reviewer 2 Report

The manuscript describes the synthesis of four-arm (PSt)4 and (PtBA)star macroinitiators using enhanced spin capturing polymerisation. These macroinitiators were subsequently chain-extended to form miktoarm star copolymers (μ-stars) of (PSt)2-μ-(PtBA-b-PSt)2 and (PtBA)2-μ-(PSt-b-PtBA)2) using NMP. The hydrolysis of the tert-butyl groups yielded the amphiphilic μ-stars of (PSt)2-μ-(PAA-b-PSt)2 and (PAA)2-μ-(PSt-b-PAA)2. The synthesized polymers were reported to have different physical properties observed via water contact angle and Atomic Force Microscope (AFM).  

The manuscript revealed an interesting approach (i.e, ESCP followed by NMP) to prepare star copolymers. Both design of experiments and presentation of data were logical and good. The authors used a wide range of methods to characterise the copolymers. I recommend that this manuscript should be published in Nanomaterials after the following major comments are addressed.

  1. Line 185-187, after hydrolysis of tert-butyl group, the mixture was dialysed against water. I wonder if the resulting star polymer was soluble in water. If not, it might trap the impurities inside during the dialysis. Do you have any comments on it?
  2. Have the authors tried to run GPC for the hydrolysed star polymer? I understand that the carboxylic group might interact with the material column and lead to the unpredicted MW distribution. However, the interaction could be eliminated by experienced researchers. The reason I mentioned this exercise is because that it is important to confirm if the star polymers remain intact after experiencing a harsh hydrolytic condition (i.e, TFA).
  3. Line 201, the authors mentioned “ESCP typically proceeds with a conventional free radical polymerization fashion”. How does this outcome compare to the previous works in the literature? The microstructure of the polymers highly depends on their dispersity. While other controlled radical polymerisation methods can produce well-defied and low dispersity polymers, the reported method gave quite dispersed polymers. This method of preparing polymer may not be suitable for tuning the microstructure of the polymer. Can you please comment on this matter?
  4. Figure 3b seemed to have something at the high molecular region (a half peak at logMW >4.6). Is it due to a coupling of the polymers? Do the authors have any comments on this?
  5. The graphs in line 211 and 265 should be improved. The line could not be seen easily and therefore it should be changed to other style and colour.
  6. In line 311, the authors did not mention the method of spin coating of the polymer on the substrates (i.e., solvent used to disperse the polymers, rotating speed, etc.) since the solvent had a significant influence on the surface of the coating resulting in changing the contact angle. Was the same solvent used for all the polymers? Is it possible to observe the microstructures of the coated substrates under electron microscopy?
  7. Regarding to water contact angle measurement, the authors should mention the pH of the water used. If possible, using water with different pHs could provide interesting results.
  8. The authors should assign peaks for the graphs in Figure S1 and S2.

Author Response

Response to the Reviewers’ comments (Manuscript ID: nanomaterials-1355637):

For reviewer 2:

The manuscript describes the synthesis of four-arm (PSt)4 and (PtBA)4 star macroinitiators using enhanced spin capturing polymerisation. These macroinitiators were subsequently chain-extended to form miktoarm star copolymers (μ-stars) of(PSt)2-µ-(PtBA-b-PSt)2 and (PtBA)2-µ-(PSt-b-PtBA)2 using NMP. The hydrolysis of the tert-butyl groups yielded the amphiphilic μ-stars of (PSt)2-µ-(PAA-b-PSt)2 and (PAA)2-µ-(PSt-b-PAA)2. The synthesized polymers were reported to have different physical properties observed via water contact angle and Atomic Force Microscope (AFM). The manuscript revealed an interesting approach (i.e, ESCP followed by NMP) to prepare star copolymers. Both design of experiments and presentation of data were logical and good. The authors used a wide range of methods to characterise the copolymers. I recommend that this manuscript should be published in Nanomaterials after the following major comments are addressed.

General response: We sincerely thank the reviewer’s affirmative remarks. Our point-by-point responses are listed as follows.

1. Line 185-187, after hydrolysis of tert-butyl group, the mixture was dialysed against water. I wonder if the resulting star polymer was soluble in water. If not, it might trap the impurities inside during the dialysis. Do you have any comments on it?

Response: We thank the reviewer’s detailed comments. The hydrolyzed copolymers did not dissolve very well in water. So we added certain amounts of DMSO to remove impurities. We are sorry for the unclear part. We added the point in the text and marked in yellow.  

2. Have the authors tried to run GPC for the hydrolysed star polymer? I understand that the carboxylic group might interact with the material column and lead to the unpredicted MW distribution. However, the interaction could be eliminated by experienced researchers. The reason I mentioned this exercise is because that it is important to confirm if the star polymers remain intact after experiencing a harsh hydrolytic condition (i.e, TFA).

Response: We thank the reviewer’s important comments. We tried to run GPC for the hydrolysed star polymer but they didn’t dissolve well which might be ascribed to the presence of PAA segments. We agree with the comment and further confirmations of how intact after experiencing a harsh hydrolytic condition are testing currently. 

3. Line 201, the authors mentioned “ESCP typically proceeds with a conventional free radical polymerization fashion”. How does this outcome compare to the previous works in the literature? The microstructure of the polymers highly depends on their dispersity. While other controlled radical polymerisation methods can produce well-defined and low dispersity polymers, the reported method gave quite dispersed polymers. This method of preparing polymer may not be suitable for tuning the microstructure of the polymer. Can you please comment on this matter?

Response: We thank the reviewer’s important comments. ESCP does not possess the feature of controlled/living radical polymerization. We addressed this point and reported literature in the text. We marked the text in yellow. 

4. Figure 3b seemed to have something at the high molecular region (a half peak at logMW >4.6). Is it due to a coupling of the polymers? Do the authors have any comments on this?

Response: We thank the reviewer’s comment. To cleave the star polymer, it won’t happen coupling reactions in the presence of an excess of TBTH. It thus might be due to the baseline issue. We addressed this point in the text.  

5. The graphs in line 211 and 265 should be improved. The line could not be seen easily and therefore it should be changed to other style and colour.

Response: We thank the reviewer’s comment. We improved the graphs with clear content. 

6. In line 311, the authors did not mention the method of spin coating of the polymer on the substrates (i.e., solvent used to disperse the polymers, rotating speed, etc.) since the solvent had a significant influence on the surface of the coating resulting in changing the contact angle. Was the same solvent used for all the polymers? Is it possible to observe the microstructures of the coated substrates under electron microscopy?

Response: We thank the reviewer’s important comment. We used the same solvents and drop-casting methods to obtain thin layers. Further various treatments and SEM measurements will be conducted in near future and hope to present soon after the tough situation. We added the details for the process and marked in yellow. 

7. Regarding to water contact angle measurement, the authors should mention the pH of the water used. If possible, using water with different pHs could provide interesting results.

Response: We thank the reviewer’s valuable comment. We used neutral DI water for testing and we added the description. Further WCA measurements with various pH water droplets are currently underway. We hope to present soon after the tough situation.

8. The authors should assign peaks for the graphs in Figure S1 and S2.

Response: We thank the reviewer’s comment. We added some representative peaks in Figs. S1 and S2. 

Reviewer 3 Report

Title: Miktoarm Star Copolymers Prepared by Transformation from Enhanced Spin Capturing Polymerizatioon to Nitroxide-Mediated Polymerization (ESCP-T-NMP) toward Nanomaterials (Submitted to nanomaterials)

   This paper demonstrates the synthesis of m-star polymer composed of PSt (or PtBA) and PSt-b-PtBA segments and evaluate the thermal properties, wettability, and morphology of before and after hydrolysis polymers. It is interesting synthetic route which can be converted from a regular star polymer to a miktoarm star by consisting of two pathways, a radical trap reaction and NMP. The paper is well written and clear, and the conclusions are supported by the data.However, the authors would need to address several issues before publication. We support the publication on this manuscript with the following minor revisions.

・ The supplemental citation of the literatures regarding m-star polymers (Chem. Rev., 2016, 116, 6743-6836, ACS Macro Lett. 2013, 2, 625-629) is suggested.

・ Concerning about the sentence “It is reported that the more complex molecular architectures ~~~ those of linear polymers in a similar MW” line 246-248 in page 7;  Indeed, branched polymers have a lower hydrodynamic radius than linear polymers, which can lead to lower molecular weights. However, it is unlikely that the GPC curve will change to the low molecular weight side after the chain extension reaction. Therefore, it should be demonstrated that the miktoarm star polymer obtained after the chain extension reaction also undergoes a reduction reaction to obtain a block polymer. In addition, it may be possible to control the molecular weight by chain extension reaction via NMP, and what is the design molecular weight of the block copolymer segment?

・ In the thermal properties, it should be better to display the results of PSt and PtBMA (and PAA) homopolymers.

・The authors predict a structure with PAA segments on the surface, but why does the contact angle change over time? The authors should perform further structural analysis of the outermost surface by XPS measurement.

・The authors should correct the word from “N-(tert-butyl) hydroxy aclamineetate” to “N-(tert-butyl) hydroxy amine”.

Author Response

Response to the Reviewers’ comments (Manuscript ID: nanomaterials-1355637):

For reviewer 3:

This paper demonstrates the synthesis of m-star polymer composed of PSt (or PtBA) and PSt-b-PtBA segments and evaluate the thermal properties, wettability, and morphology of before and after hydrolysis polymers. It is interesting synthetic route which can be converted from a regular star polymer to a miktoarm star by consisting of two pathways, a radical trap reaction and NMP. The paper is well written and clear, and the conclusions are supported by the data. However, the authors would need to address several issues before publication. We support the publication on this manuscript with the following minor revisions.

General response: We sincerely thank the reviewer’s positive remarks. Our point-by-point responses are listed as follows.

1. The supplemental citation of the literatures regarding m-star polymers (Chem. Rev., 2016, 116, 6743-6836, ACS Macro Lett. 2013, 2, 625-629) is suggested.

Response: We thank the reviewer’s valuable suggestion. We added the literature and marked in yellow.

2. Concerning about the sentence “It is reported that the more complex molecular architectures ~~~ those of linear polymers in a similar MW” line 246-248 in page 7; Indeed, branched polymers have a lower hydrodynamic radius than linear polymers, which can lead to lower molecular weights. However, it is unlikely that the GPC curve will change to the low molecular weight side after the chain extension reaction. Therefore, it should be demonstrated that the miktoarm star polymer obtained after the chain extension reaction also undergoes a reduction reaction to obtain a block polymer. In addition, it may be possible to control the molecular weight by chain extension reaction via NMP, and what is the design molecular weight of the block copolymer segment?

Response: We thank the reviewer’s critical comments. We agree with our abnormal observations in GPC traces and it possibly undergoes a reduction reaction. However, a similar chemical structure of mono-functional nitrone was reported by Barner-Kowollik group. Its effectiveness of chain extension via NMP was demonstrated. We thus emphasized another possible reason that it could be due to the insertion of polymer chains during NMP leading to m-star copolymers with an asymmetric molecular conformation. We added this possible reason about the results and corrected our design MW of BCP segment information and marked in yellow. 

3. In the thermal properties, it should be better to display the results of PSt and PtBA (and PAA) homopolymers.

Response: We thank the reviewer’s important comment. We added the TGA traces of linear homopolymers in Fig. 8. 

4. The authors predict a structure with PAA segments on the surface, but why does the contact angle change over time? The authors should perform further structural analysis of the outermost surface by XPS measurement.

Response: We thank the reviewer’s important comment. The WCA changes were due to the surficial swelling behaviors. But unfortunately, we can’t assess to measure XPS in a few months. Again, further systematic measurements, e.g., SEM, TEM, and XPS are currently underway. We hope to present soon after the tough situation. We addressed the point about WCA changes in the text and added a reported study. 

5. The authors should correct the word from “N-(tert-butyl) hydroxy aclamineetate” to “N-(tert-butyl) hydroxy amine”.

Response: We thank the reviewer’s check. We revised the typo and marked in yellow. 

Reviewer 4 Report

The manuscript entitled “Miktoarm Star Copolymers Prepared by Transformation from Enhanced Spin Capturing Polymerization to Nitroxide-Mediated Polymerization (ESCP-Ŧ-NMP) toward Nanomaterials” describes the synthesis of four-arm miktoarm star polymers (PSt)2-µ-(PtBA-b-PSt)2 and (PtBA)2-µ-(PSt-b-PtBA)2 where PtBA denotes poly(tert-butyl acrylate) and PSt denotes polystyrene. Hydrolysis was subsequently performed on the PtBA segments to convert them into poly(acrylic acid)(PAA), thus yielding (PSt)2-μ-(PAA-b-PSt)2 and (PAA)2-μ-(PSt-b-PAA)2. These polymers were characterized via various methods such as thermogravimetric analysis (TGA), differential scanning calorimetry (DSC), atomic force microscopy (AFM), 1H NMR spectroscopy, FT-IR spectroscopy, and water contact angle (WCA) measurements. These polymers were prepared by a facile polymerization transformation approach.
    Overall, the procedure used to prepare these miktoarm copolymers is innovative and the results are well-supported by a good range of experimental data. With regard to the WCAs, it appears that static contact angles were obtained. Possibly it may be helpful to supplement these static contact angles with advancing and receding contact angles, or possibly sliding angle measurements, although I do not believe this is a critical issue as the contact angles were not the primary emphasis of this study.
    The manuscript is generally well-written, but could benefit from some minor polishing, and some minor suggestions in this regard are provided below.
    I believe that this work will be is significant interest to the readers to the readers of Nanomaterials, and this work has relevance in a wide range of areas, including synthetic chemistry, architectural polymers, materials science, and self-assembly. I believe that this work is suitable for publication pending minor revisions, which are outlined below.

Line 31: “each step of polymerization” can be changed to “each step of the polymerization” or “each polymerization step”.

Lines 47-48: “and a numerous of others” can be changed to “and numerous other methods” or “and a number of other methods”.

Line 58: “macromolecular” can be changed to “macromolecules” or possibly “macromolecular systems”.

Lines 60-61: The phrase “are typically belonged to the analog of linear type copolymers” is a little unclear.

Line 65: “has paid great attention” can be changed to “has received great attention” or “has received significant attention” or “has received much attention”.

Line 74: “for industrials” can be changed to “for industrial applications” or “for industrial processes”.

Line 78: “NMP technique” can be changed to “the NMP technique”.

Line 86: “quickly go cross-coupling” can possibly be changed to “quickly undergo cross-coupling”.

Line 87: The phrase “subsequently go for NMP conducting” seems to be a little unclear.

Lines 92-93: “The beauty of this convergent approach can change the” can possibly be changed to “The beauty of this convergent approach is that it can change the” or “The beauty of this convergent approach is that it can be used to change the”.

Line 104: “can be also attained” can possibly be changed to “can also be attained”.

Line 131: “Bruker 400 NMR spectrometer” can be changed to “A Bruker 400 NMR spectrometer”.

Lines 144 and 145: The significant figures (sig. figs.) of the masses (in g) do not seem to match the significant figures of the corresponding mmol values for the number sets “(1 g, 7.46 mmol)” (as there is 1 sig. fig. shown for the g value and 3 for the mmol value) and “(2.4 g, 16.4 mmol)” (as there are 2 sig. figs. for the g value but 3 for the mmol value).

Line 149: “The crude was” can be changed to “The crude product was”.

Line 157: “and few drops of” can be changed to “and a few drops of”.

Lines 168: The significant figures in the set of numbers “(0.62 mL, 2 mmol)” do not match, as there are 2 sig. figs. for the mL value but 1 for the mmol value.

Lines 173-174: “scheme 2” should be changed to “Scheme 2”.

Line 174: The significant figures in the set of numbers “(0.82 g, 0.03 mmol)” do not match (there are 2 sig. figs for the g value but 1 for the mmol value).

Line 184: The significant figures in the set of numbers “(0.72 g, 0.06 mmol)” do not match (there are 2 sig. figs for the g value but 1 for the mmol value).

Line 201: “indicated poor-controlled polymerizations and high PDI values” can be changed to “indicated that the polymerizations were poorly-controlled and that high PDI values were obtained”.

Lines 207-208, Figure 1: Error mars may be needed for the data points in Figure 1.

Line 222: The phrase “and generations indivisible” is a little unclear.

Lines 236, 237 and 239: Error margins may be needed for the reaction rates.

Line 248: “environment comparing to those of” can possibly be changed to “environment in comparison to those of”.

Lines 262, 263: Error bars may be needed for the date points in Figure 4.

Line 274: “obtaining products after conducting the hydrolysis reactions” can be changed to “products that were obtained after the hydrolysis reactions had been performed” or “products that were obtained after the hydrolysis reactions were completed”.

Lines 313, 317, 319, and 323, and 325: error margins may be needed for the water contact angle measurements.

Lines 339-340: possibly “less-hard PSt chains” can be changed to “softer PSt chains”.

Author Response

Response to the Reviewers’ comments (Manuscript ID: nanomaterials-1355637):

For reviewer 4:

The manuscript entitled “Miktoarm Star Copolymers Prepared by Transformation from Enhanced Spin Capturing Polymerization to Nitroxide-Mediated Polymerization (ESCP-Ŧ-NMP) toward Nanomaterials” describes the synthesis of four-arm miktoarm star polymers (PSt)2-µ-(PtBA-b-PSt)2 and (PtBA)2-µ-(PSt-b-PtBA)2 where PtBA denotes poly(tert-butyl acrylate) and PSt denotes polystyrene. Hydrolysis was subsequently performed on the PtBA segments to convert them into poly(acrylic acid)(PAA), thus yielding (PSt)2-µ-(PAA-b-PSt)2 and (PAA)2-µ-(PSt-b-PAA)2. These polymers were characterized via various methods such as thermogravimetric analysis (TGA), differential scanning calorimetry (DSC), atomic force microscopy (AFM), 1H NMR spectroscopy, FT-IR spectroscopy, and water contact angle (WCA) measurements. These polymers were prepared by a facile polymerization transformation approach. Overall, the procedure used to prepare these miktoarm copolymers is innovative and the results are well-supported by a good range of experimental data. With regard to the WCAs, it appears that static contact angles were obtained. Possibly it may be helpful to supplement these static contact angles with advancing and receding contact angles, or possibly sliding angle measurements, although I do not believe this is a critical issue as the contact angles were not the primary emphasis of this study. The manuscript is generally well-written, but could benefit from some minor polishing, and some minor suggestions in this regard are provided below. I believe that this work will be is significant interest to the readers to the readers of Nanomaterials, and this work has relevance in a wide range of areas, including synthetic chemistry, architectural polymers, materials science, and self-assembly. I believe that this work is suitable for publication pending minor revisions, which are outlined below. 

General response: We sincerely thank the reviewer’s positive remarks. Our responses are listed as follows.

1. Line 31: “each step of polymerization” can be changed to “each step of the polymerization” or “each polymerization step”.

2. Lines 47-48: “and a numerous of others” can be changed to “and numerous other methods” or “and a number of other methods”.

3. Line 58: “macromolecular” can be changed to “macromolecules” or possibly “macromolecular systems”.

4. Lines 60-61: The phrase “are typically belonged to the analog of linear type copolymers” is a little unclear.

5. Line 65: “has paid great attention” can be changed to “has received great attention” or “has received significant attention” or “has received much attention”.

6. Line 74: “for industrials” can be changed to “for industrial applications” or “for industrial processes”.

7. Line 78: “NMP technique” can be changed to “the NMP technique”.

8. Line 86: “quickly go cross-coupling” can possibly be changed to “quickly undergo cross-coupling”.

9. Line 87: The phrase “subsequently go for NMP conducting” seems to be a little unclear.

10. Lines 92-93: “The beauty of this convergent approach can change the” can possibly be changed to “The beauty of this convergent approach is that it can change the” or “The beauty of this convergent approach is that it can be used to change the”.

11. Line 104: “can be also attained” can possibly be changed to “can also be attained”.

12. Line 131: “Bruker 400 NMR spectrometer” can be changed to “A Bruker 400 NMR spectrometer”.

13. Lines 144 and 145: The significant figures (sig. figs.) of the masses (in g) do not seem to match the significant figures of the corresponding mmol values for the number sets “(1 g, 7.46 mmol)” (as there is 1 sig. fig. shown for the g value and 3 for the mmol value) and “(2.4 g, 16.4 mmol)” (as there are 2 sig. figs. for the g value but 3 for the mmol value).

14. Line 149: “The crude was” can be changed to “The crude product was”.

15. Line 157: “and few drops of” can be changed to “and a few drops of”.

16. Lines 168: The significant figures in the set of numbers “(0.62 mL, 2 mmol)” do not match, as there are 2 sig. figs. for the mL value but 1 for the mmol value.

17. Lines 173-174: “scheme 2” should be changed to “Scheme 2”.

18. Line 174: The significant figures in the set of numbers “(0.82 g, 0.03 mmol)” do not match (there are 2 sig. figs for the g value but 1 for the mmol value).

19. Line 184: The significant figures in the set of numbers “(0.72 g, 0.06 mmol)” do not match (there are 2 sig. figs for the g value but 1 for the mmol value).

20. Line 201: “indicated poor-controlled polymerizations and high PDI values” can be changed to “indicated that the polymerizations were poorly-controlled and that high PDI values were obtained”.

21. Lines 207-208, Figure 1: Error mars may be needed for the data points in Figure 1.

22. Line 222: The phrase “and generations indivisible” is a little unclear.

23. Lines 236, 237 and 239: Error margins may be needed for the reaction rates.

24. Line 248: “environment comparing to those of” can possibly be changed to “environment in comparison to those of”.

25. Lines 262, 263: Error bars may be needed for the date points in Figure 4.

26. Line 274: “obtaining products after conducting the hydrolysis reactions” can be changed to “products that were obtained after the hydrolysis reactions had been performed” or “products that were obtained after the hydrolysis reactions were completed”.

27. Lines 313, 317, 319, and 323, and 325: error margins may be needed for the water contact angle measurements.

28. Lines 339-340: possibly “less-hard PSt chains” can be changed to “softer PSt chains”.

Response: We sincerely thank the reviewer’s careful and detailed checking. The abovementioned comments were all revised, including typos, descriptions, significant figures, error bars, and error margins in both text and figures. We marked all the changes in yellow. 

Round 2

Reviewer 1 Report

With the revisions, the authors addressed the major concerns I had.

Reviewer 2 Report

I recommend that this manuscript is published in the current form.